# Adaptive Clustering through Semidefinite Programming

**Martin Royer**

Laboratoire de Mathématiques d'Orsay, Univ. Paris-Sud, CNRS,
Université Paris-Saclay,
91405 Orsay, France
`martin.royer@math.u-psud.fr`

## Abstract

We analyze the clustering problem through a flexible probabilistic model that aims to identify an optimal partition on the sample $X_1, ..., X_n$. We perform exact clustering with high probability using a convex semidefinite estimator that interprets as a corrected, relaxed version of $K$-means. The estimator is analyzed through a non-asymptotic framework and showed to be optimal or near-optimal in recovering the partition. Furthermore, its performances are shown to be adaptive to the problem's effective dimension, as well as to $K$ the unknown number of groups in this partition. We illustrate the method's performances in comparison to other classical clustering algorithms with numerical experiments on simulated high-dimensional data.

## 1 Introduction

Clustering, a form of unsupervised learning, is the classical problem of assembling $n$ observations $X_1, ..., X_n$ from a $p$-dimensional space into $K$ groups. Applied fields are craving for robust clustering techniques, such as computational biology with genome classification, data mining or image segmentation from computer vision. But the clustering problem has proven notoriously hard when the embedding dimension is large compared to the number of observations (see for instance the recent discussions from [2, 21]).

A famous early approach to clustering is to solve for the geometrical estimator K-means [19, 13, 14]. The intuition behind its objective is that groups are to be determined in a way to minimize the total intra-group variance. It can be interpreted as an attempt to "best" represent the observations by $K$ points, a form of vector quantization. Although the method shows great performances when observations are homoscedastic, K-means is a NP-hard, ad-hoc method. Clustering with probabilistic frameworks are usually based on maximum likelihood approaches paired with a variant of the EM algorithm for model estimation, see for instance the works of Fraley & Raftery [11] and Dasgupta & Schulman [9]. These methods are widespread and popular, but they tend to be very sensitive to initialization and model misspecifications.

Several recent developments establish a link between clustering and semidefinite programming. Peng & Wei [17] show that the K-means objective can be relaxed into a convex, semidefinite program, leading Mixon *et al.* [16] to use this relaxation under a subgaussian mixture model to estimate the cluster centers. Yan and Sarkar [24] use a similar semidefinite program in the context of covariate clustering, when the network has nodes and covariates. Chrétien *et al.* [8] use a slightly different form of a semidefinite program to recover the adjacency matrix of the cluster graph with high probability. Lastly in the different context of variable clustering, Bunea *et al.* [6] present a semidefinite program with a correction step to produce non-asymptotic exact recovery results.

In this work, we build upon the work and context of [6], and transpose and adapt their ideas for point clustering: we introduce a semidefinite estimator for point clustering inspired by the findings of [17] with a correction component originally presented in [6]. We show that it produces a very strong contender for clustering recovery in terms of speed, adaptivity and robustness to model perturbations. In order to do so we produce a flexible probabilistic model inducing an optimal partition of the data that we aim to recover. Using the same structure of proof in a different context, we establish elements of stochastic control (see for instance Lemma A.1 on the concentration of random subgaussian Gram matrices in the supplementary material) to derive conditions of exact clustering recovery with high probability and show optimal performances – including in high dimensions, improving on [16], as well as adaptivity to the effective dimension of the problem. We also show that our results continue to hold without knowledge of the number of structures given one single positive tuning parameter. Lastly we provide evidence of our method's efficiency and further insight from simulated data.

**Notation.** Throughout this work we use the convention $0/0 := 0$ and $[n] = \{1, ..., n\}$. We take $a_n \lesssim b_n$ to mean that $a_n$ is smaller than $b_n$ up to an absolute constant factor. Let $\mathcal{S}_{d-1}$ denote the unit sphere in $\mathbb{R}^d$. For $q \in \mathbb{N}^* \cup \{+\infty\}$, $\nu \in \mathbb{R}^d$, $|\nu|_q$ is the $l_q$-norm and for $M \in \mathbb{R}^{d \times d'}$, $|M|_q$, $|M|_F$ and $|M|_{op}$ are respectively the entry-wise $l_q$-norm, the Frobenius norm associated with scalar product $\langle ., . \rangle$ and the operator norm. $|D|_V$ is the variation semi-norm for a diagonal matrix $D$, the difference between its maximum and minimum element. Let $A \succcurlyeq B$ mean that $A - B$ is symmetric, positive semidefinite.

## 2 Probabilistic modeling of point clustering

Consider $X_1, ..., X_n$ and let $\nu_a = \mathbb{E}[X_a]$. The variable $X_a$ can be decomposed into

$$X_a = \nu_a + E_a, \quad a = 1, ..., n, \tag{1}$$

with $E_a$ stochastic centered variables in $\mathbb{R}^p$.

**Definition 1.** *For $K > 1$, $\boldsymbol{\mu} = (\mu_1, ..., \mu_K) \in (\mathbb{R}^p)^K$, $\delta \geqslant 0$ and $\mathcal{G} = \{G_1, ..., G_K\}$ a partition of $[n]$, we say $X_1, ..., X_n$ are $(\mathcal{G}, \boldsymbol{\mu}, \delta)$-clustered if $\forall k \in [K], \forall a \in G_k, |\nu_a - \mu_k|_2 \leqslant \delta$. We then call*

$$\Delta(\boldsymbol{\mu}) := \min_{k < l} |\mu_k - \mu_l|_2 \tag{2}$$

*the separation between the cluster means, and*

$$\rho(\mathcal{G}, \boldsymbol{\mu}, \delta) := \Delta(\boldsymbol{\mu})/\delta \tag{3}$$

*the discriminating capacity of $(\mathcal{G}, \boldsymbol{\mu}, \delta)$.*

In this work we assume that $X_1, ..., X_n$ are $(\mathcal{G}, \boldsymbol{\mu}, \delta)$-clustered. Notice that this definition does not impose any constraint on the data: for any given $\mathcal{G}$, there exists a choice of $\boldsymbol{\mu}$, means and radius $\delta$ important enough so that $X_1, ..., X_n$ are $(\mathcal{G}, \boldsymbol{\mu}, \delta)$-clustered. But we are interested in partitions with greater discriminating capacity, i.e. that make more sense in terms of group separation. Indeed remark that if $\rho(\mathcal{G}, \boldsymbol{\mu}, \delta) < 2$, the population clusters $\{\nu_a\}_{a \in G_1}, ..., \{\nu_a\}_{a \in G_K}$ are not linearly separable, but a high $\rho(\mathcal{G}, \boldsymbol{\mu}, \delta)$ implies that they are well-separated from each other. Furthermore, we have the following result.

**Proposition 1.** *Let $(\mathcal{G}_K^*, \boldsymbol{\mu}^*, \delta^*) \in \arg\max \rho(\mathcal{G}, \boldsymbol{\mu}, \delta)$ for $(\mathcal{G}, \boldsymbol{\mu}, \delta)$ such that $X_1, ..., X_n$ are $(\mathcal{G}, \boldsymbol{\mu}, \delta)$-clustered, and $|\mathcal{G}| = K$. If $\rho(\mathcal{G}_K^*, \boldsymbol{\mu}^*, \delta^*) > 4$ then $\mathcal{G}_K^*$ is the unique maximizer of $\rho(\mathcal{G}, \boldsymbol{\mu}, \delta)$.*

So $\mathcal{G}_K^*$ is the partition maximizing the discriminating capacity over partitions of size $K$. Therefore in this work, we will assume that there is a $K > 1$ such that $X_1, ..., X_n$ is $(\mathcal{G}, \boldsymbol{\mu}, \delta)$-clustered with $|\mathcal{G}| = K$ and $\rho(\mathcal{G}, \boldsymbol{\mu}, \delta) > 4$. By Proposition 1, $\mathcal{G}$ is then identifiable. It is the partition we aim to recover.

We also assume that $X_1, ..., X_n$ are independent observations with subgaussian behavior. Instead of the classical isotropic definition of a subgaussian random vector (see for example [20]), we use a more flexible definition that can account for anisotropy.

**Definition 2.** *Let $Y$ be a random vector in $\mathbb{R}^d$, $Y$ has a subgaussian distribution if there exist $\Sigma \in \mathbb{R}^{d \times d}$ such that $\forall x \in \mathbb{R}^d$,*

$$\mathbb{E}\left[e^{x^T(Y - \mathbb{E}Y)}\right] \leqslant e^{x^T \Sigma x/2}. \tag{4}$$

We then call $\Sigma$ a variance-bounding matrix of random vector $Y$, and write shorthand $Y \sim \text{subg}(\Sigma)$. Note that $Y \sim \text{subg}(\Sigma)$ implies $\text{Cov}(Y) \preccurlyeq \Sigma$ in the semidefinite sense of the inequality. To sum-up our modeling assumptions in this work:

**Hypothesis 1.** *Let $X_1, ..., X_n$ be independent, subgaussian, $(\mathcal{G}, \boldsymbol{\mu}, \delta)$-clustered with $\rho(\mathcal{G}, \boldsymbol{\mu}, \delta) > 4$.*

Remark that the modelization of Hypothesis 1 can be connected to another popular probabilistic model: if we further ask that $X_1, ..., X_n$ are identically-distributed within a group (and hence $\delta = 0$), the model becomes a realization of a *mixture model*.

## 3 Exact partition recovery with high probability

Let $\mathcal{G} = \{G_1, ..., G_K\}$ and $m := \min_{k \in [K]} |G_k|$ denote the minimum cluster size. $\mathcal{G}$ can be represented by its caracteristic matrix $B^* \in \mathbb{R}^{n \times n}$ defined as $\forall k, l \in [K]^2, \forall (a, b) \in G_k \times G_l$,

$$B^*_{ab} := \begin{cases} 1/|G_k| & \text{if } k = l \\ 0 & \text{otherwise.} \end{cases}$$

In what follows, we will demonstrate the recovery of $\mathcal{G}$ through recovering its caracteristic matrix $B^*$. We introduce the sets of square matrices

$$\mathcal{C}_K^{\{0,1\}} := \{B \in \mathbb{R}_+^{n \times n} : B^T = B, \text{tr}(B) = K, B 1_n = 1_n, B^2 = B\} \tag{5}$$

$$\mathcal{C}_K := \{B \in \mathbb{R}_+^{n \times n} : B^T = B, \text{tr}(B) = K, B 1_n = 1_n, B \succcurlyeq 0\} \tag{6}$$

$$\mathcal{C} := \bigcup_{K \in \mathbb{N}} \mathcal{C}_K. \tag{7}$$

We have: $\mathcal{C}_K^{\{0,1\}} \subset \mathcal{C}_K \subset \mathcal{C}$ and $\mathcal{C}_K$ is convex. Notice that $B^* \in \mathcal{C}_K^{\{0,1\}}$. A result by Peng, Wei (2007) [17] shows that the K-means estimator $\bar{B}$ can be expressed as

$$\bar{B} = \underset{B \in \mathcal{C}_K^{\{0,1\}}}{\arg\max} \langle \widehat{\Lambda}, B \rangle \tag{8}$$

for $\widehat{\Lambda} := (\langle X_a, X_b \rangle)_{(a,b) \in [n]^2} \in \mathbb{R}^{n \times n}$, the observed Gram matrix. Therefore a natural relaxation is to consider the following estimator:

$$\widehat{B} := \underset{B \in \mathcal{C}_K}{\arg\max} \langle \widehat{\Lambda}, B \rangle. \tag{9}$$

Notice that $\mathbb{E}\, \widehat{\Lambda} = \Lambda + \Gamma$ for $\Lambda := (\langle \nu_a, \nu_b \rangle)_{(a,b) \in [n]^2} \in \mathbb{R}^{n \times n}$, and $\Gamma := \mathbb{E}\left[ \langle E_a, E_b \rangle \right]_{(a,b) \in [n]^2} = \text{diag}\left(\text{tr}(\text{Var}(E_a))\right)_{1 \leqslant a \leqslant n} \in \mathbb{R}^{n \times n}$. The following two results demonstrate that $\Lambda$ is the signal structure that lead the optimizations of (8) and (9) to recover $B^*$, whereas $\Gamma$ is a bias term that can hurt the process of recovery.

**Proposition 2.** *There exist $c_0 > 1$ absolute constant such that if $\rho^2(\mathcal{G}, \boldsymbol{\mu}, \delta) > c_0(6 + \sqrt{n}/m)$ and $m\Delta^2(\boldsymbol{\mu}) > 8|\Gamma|_V$, then we have*

$$\underset{B \in \mathcal{C}_K^{\{0,1\}}}{\arg\max} \langle \Lambda + \Gamma, B \rangle = B^* = \underset{B \in \mathcal{C}_K}{\arg\max} \langle \Lambda + \Gamma, B \rangle. \tag{10}$$

This proposition shows that the $\widehat{B}$ estimator, as well as the K-means estimator, would recover partition $\mathcal{G}$ on the population Gram matrix if the variation semi-norm of $\Gamma$ were sufficiently small compared to the cluster separation. Notice that to recover the partition on the population version, we require the discriminating capacity to grow as fast as $1 + (\sqrt{n}/m)^{1/2}$ instead of simply 1 from Hypothesis 1. The following proposition demonstrates that if the condition on the variation semi-norm of $\Gamma$ is not met, $\mathcal{G}$ may not even be recovered on the population version.

**Proposition 3.** *There exist $\mathcal{G}, \boldsymbol{\mu}, \delta$ and $\Gamma$ such that $\rho^2(\mathcal{G}, \boldsymbol{\mu}, \delta) = +\infty$ but we have $m\Delta^2(\boldsymbol{\mu}) < 2|\Gamma|_V$ and*

$$B^* \notin \underset{B \in \mathcal{C}_K^{\{0,1\}}}{\arg\max} \langle \Lambda + \Gamma, B \rangle \quad \text{and} \quad B^* \notin \underset{B \in \mathcal{C}_K}{\arg\max} \langle \Lambda + \Gamma, B \rangle. \tag{11}$$

So Proposition 3 shows that even if the population clusters are perfectly discriminated, there is a configuration for the variances of the noise that makes it impossible to recover the right clustering by K-means. This shows that K-means may fail when the random variable homoscedasticity assumption is violated, and that it is important to correct for $\Gamma = \mathrm{diag}(\mathrm{tr}(\mathrm{Var}(E_a)))_{1 \leqslant a \leqslant n}$.

Suppose we produce such an estimator $\widehat{\Gamma}^{corr}$. Then substracting $\widehat{\Gamma}^{corr}$ from $\widehat{\Lambda}$ can be interpreted as a correcting term, i.e. a way to de-bias $\widehat{\Lambda}$ as an estimator of $\Lambda$. Hence the previous results demonstrate the interest of studying the following semi-definite estimator of the projection matrix $B^*$, let

$$\widehat{B}^{corr} := \underset{B \in \mathcal{C}_K}{\arg\max} \langle \widehat{\Lambda} - \widehat{\Gamma}^{corr}, B \rangle. \tag{12}$$

In order to demonstrate the recovery of $B^*$ by this estimator, we introduce different quantitative measures of the "spread" of our stochastic variables, that affect the quality of the recovery. By Hypothesis 1 there exist $\Sigma_1, ..., \Sigma_n$ such that $\forall a \in [n]$, $X_a \sim \mathrm{subg}(\Sigma_a)$. Let

$$\sigma^2 := \max_{a \in [n]} |\Sigma_a|_{op}, \quad \mathcal{V}^2 := \max_{a \in [n]} |\Sigma_a|_F, \quad \gamma^2 := \max_{a \in [n]} \mathrm{tr}(\Sigma_a) \tag{13}$$

We now produce $\widehat{\Gamma}^{corr}$. Since there is no relation between the variances of the points in our model, there is very little hope of estimating $\mathrm{Var}(E_a)$. As for our quantity of interest $\mathrm{tr}(\mathrm{Var}(E_a))$, a form of volume, a rough estimation is challenging but possible. The estimator from [6] can be adapted to our context. For $(a,b) \in [n]^2$ let $V(a,b) := \max_{(c,d) \in ([n] \backslash \{a,b\})^2} \left| \langle X_a - X_b, \frac{X_c - X_d}{|X_c - X_d|_2} \rangle \right|$, $\widehat{b}_1 := \arg\min_{b \in [n] \backslash \{a\}} V(a,b)$ and $\widehat{b}_2 := \arg\min_{b \in [n] \backslash \{a, \widehat{b}_1\}} V(a,b)$. Then for $a \in [n]$, let

$$\widehat{\Gamma}^{corr} := \mathrm{diag}\left( \langle X_a - X_{\widehat{b}_1}, X_a - X_{\widehat{b}_2} \rangle_{a \in [n]} \right). \tag{14}$$

**Proposition 4.** *Assume that $m > 2$. For $c_6, c_7 > 0$ absolute constants, with probability larger than $1 - c_6/n$ we have*

$$|\widehat{\Gamma}^{corr} - \Gamma|_\infty \leqslant c_7 \left( \sigma^2 \log n + (\delta + \sigma\sqrt{\log n})\gamma + \delta^2 \right). \tag{15}$$

So apart from the radius $\delta$ terms, that come from generous model assumptions, a proxy for $\Gamma$ is produced at a $\sigma^2 \log n$ rate that we could not expect to improve on. Nevertheless, this control on $\Gamma$ is key to attain the optimal rates below. It is general and completely independent of the structure of $\mathcal{G}$, as there is no relation between $\mathcal{G}$ and $\Gamma$.

We are now ready to introduce this paper's main result: a condition on the separation between the cluster means sufficient for ensuring recovery of $B^*$ with high probability.

**Theorem 1.** *Assume that $m > 2$. For $c_1, c_2 > 0$ absolute constants, if*

$$m\Delta^2(\boldsymbol{\mu}) \geqslant c_2 \left( \sigma^2(n + m\log n) + \mathcal{V}^2(\sqrt{n + m\log n}) + \gamma(\sigma\sqrt{\log n} + \delta) + \delta^2(\sqrt{n} + m) \right), \tag{16}$$

*then with probability larger than $1 - c_1/n$ we have $\widehat{B}^{corr} = B^*$, and therefore $\widehat{\mathcal{G}}^{corr} = \mathcal{G}$.*

We call the right hand side of (16) the separating rate. Notice that we can read two kinds of requirements coming from the separating rate: requirements on the radius $\delta$, and requirements on $\sigma^2, \mathcal{V}^2, \gamma$ dependent on the distributions of observations. It appears as if $\delta + \sigma\sqrt{\log n}$ can be interpreted as a geometrical width of our problem. If we ask that $\delta$ is of the same order as $\sigma\sqrt{\log n}$, a maximum gaussian deviation for $n$ variables, then all conditions on $\delta$ from (16) can be removed. Thus for convenience of the following discussion we will now assume $\delta \lesssim \sigma\sqrt{\log n}$.

How optimal is the result from Theorem 1? Notice that our result is adapted to anisotropy in the noise, but to discuss optimality it is easier to look at the isotropic scenario: $\mathcal{V}^2 = \sqrt{p}\sigma^2$ and $\gamma^2 = p\sigma^2$. Therefore $\Delta^2(\boldsymbol{\mu})/\sigma^2$ represents a signal-to-noise ratio. For simplicity let us also assume that all groups have equal size, that is $|G_1| = ... = |G_K| = m$ so that $n = mK$ and the sufficient condition (16) becomes

$$\frac{\Delta^2(\boldsymbol{\mu})}{\sigma^2} \gtrsim (K + \log n) + \sqrt{(K + \log n)\frac{pK}{n}}. \tag{17}$$

**Optimality.** To discuss optimality, we distinguish between low and high dimensional setups.

In the low-dimensional setup $n \vee m \log n \gtrsim p$, we obtain the following condition:

$$\frac{\Delta^2(\boldsymbol{\mu})}{\sigma^2} \gtrsim (K + \log n). \tag{18}$$

Discriminating with high probability between $n$ observations from two gaussians in dimension 1 would require a separating rate of at least $\sigma^2 \log n$. This implies that when $K \lesssim \log n$, our result is minimax. Otherwise, to our knowledge the best clustering result on approximating mixture center is from [16], and on the condition that $\Delta^2(\boldsymbol{\mu})/\sigma^2 \gtrsim K^2$. Furthermore, the $K \gtrsim \log n$ regime is known in the stochastic-block-model community as a hard regime where a gap is surmised to exist between the minimal information-theoretic rate and the minimal achievable computational rate (see for example [7]).

In the high-dimensional setup $n \vee m \log n \lesssim p$, condition (17) becomes:

$$\frac{\Delta^2(\boldsymbol{\mu})}{\sigma^2} \gtrsim \sqrt{(K + \log n) \frac{pK}{n}}. \tag{19}$$

There are few information-theoretic bounds for high-dimension clustering. Recently, Banks, Moore, Vershynin, Verzelen and Xu (2017) [3] proved a lower bound for Gaussian mixture clustering detection, namely they require a separation of order $\sqrt{K(\log K)p/n}$. When $K \lesssim \log n$, our condition is only different in that it replaces $\log(K)$ by $\log(n)$, a price to pay for going from detecting the clusters to exactly recovering the clusters. Otherwise when $K$ grows faster than $\log n$ there might exist a gap between the minimal possible rate and the achievable, as discussed previously.

**Adaptation to effective dimension.** We can analyse further the condition (16) by introducing an effective dimension $r_*$, measuring the largest volume repartition for our variance-bounding matrices $\Sigma_1, ..., \Sigma_n$. We will show that our estimator adapts to this effective dimension. Let

$$r_* := \frac{\gamma^2}{\sigma^2} = \frac{\max_{a \in [n]} \operatorname{tr}(\Sigma_a)}{\max_{a \in [n]} |\Sigma_a|_{op}}, \tag{20}$$

$r_*$ can also be interpreted as a form of global effective rank of matrices $\Sigma_a$. Indeed, define $Re(\Sigma) := \operatorname{tr}(\Sigma)/|\Sigma|_{op}$, then we have $r_* \leqslant \max_{a \in [n]} Re(\Sigma_a) \leqslant \max_{a \in [n]} \operatorname{rank}(\Sigma_a) \leqslant p$.

Now using $\mathcal{V}^2 \leqslant \sqrt{r_*}\sigma^2$ and $\gamma = \sqrt{r_*}\sigma$, condition (16) can be written as

$$\frac{\Delta^2(\boldsymbol{\mu})}{\sigma^2} \gtrsim (K + \log n) + \sqrt{(K + \log n) \frac{r_* K}{n}}. \tag{21}$$

By comparing this equation to (17), notice that $r_*$ is in place of $p$, indeed playing the role of an effective dimension for the problem. This shows that our estimator adapts to this effective dimension, without the use of any dimension reduction step. In consequence, equation (21) distinguishes between an actual high-dimensional setup: $n \vee m \log n \lesssim r_*$ and a "low" dimensional setup $r_* \lesssim n \vee m \log n$ under which, regardless of the actual value of $p$, our estimators recovers under the near-minimax condition of (18).

This informs on the effect of correcting term $\widehat{\Gamma}^{corr}$ in the theorem above when $n + m \log n \lesssim r_*$. The un-corrected version of the semi-definite program (9) has a leading separating rate of $\gamma^2/m = \sigma^2 r_*/m$, but with the $\widehat{\Gamma}^{corr}$ correction on the other hand, (21) has leading separating factor smaller than $\sigma^2 \sqrt{(K + \log n) r_*}/m = \sigma^2 \sqrt{n + m \log n} \times \sqrt{r_*}/m$. This proves that in a high-dimensional setup, our correction enhances the separating rate of at least a factor $\sqrt{(n + m \log n)/r_*}$.

## 4 Adaptation to the unknown number of group $K$

It is rarely the case that $K$ is known, but we can proceed without it. We produce an estimator adaptive to the number of groups $K$: let $\widehat{\kappa} \in \mathbb{R}_+$, we now study the following adaptive estimator:

$$\widetilde{B}^{corr} := \underset{B \in \mathcal{C}}{\arg\max} \langle \widehat{\Lambda} - \widehat{\Gamma}^{corr}, B \rangle - \widehat{\kappa} \operatorname{tr}(B). \tag{22}$$

**Theorem 2.** *Suppose that $m > 2$ and (16) is satisfied. For $c_3, c_4, c_5 > 0$ absolute constants suppose that the following condition on $\widehat{\kappa}$ is satisfied*

$$c_4\left(\mathcal{V}^2\sqrt{n} + \sigma^2 n + \gamma(\sigma\sqrt{\log n} + \delta) + \delta^2\sqrt{n}\right) < c_5\widehat{\kappa} < m\Delta^2(\boldsymbol{\mu}), \qquad (23)$$

*then we have $\widetilde{B}^{corr} = B^*$ with probability larger than $1 - c_3/n$*

Notice that condition (23) essentially requires $\widehat{\kappa}$ to be seated between $m\Delta^2(\boldsymbol{\mu})$ and some components of the right-hand side of (16). So under (23), the results from the previous section apply to the adaptive estimator $\widetilde{B}^{corr}$ as well and this shows that it is not necessary to know $K$ in order to perform well for recovering $\mathcal{G}$. Finding an optimized, data-driven parameter $\widehat{\kappa}$ using some form of cross-validation is outside of the scope of this paper.

## 5 Numerical experiments

We illustrate our method on simulated Gaussian data in two challenging, high-dimensional setup experiments for comparing clustering estimators. Our sample of $n = 100$ points are drawn from $K = 5$ identically-sized, perfectly discriminated non-isovolumic clusters of Gaussians - that is we have $\forall k \in [K], \forall a \in G_k, E_a \sim \mathcal{N}(0, \Sigma_k)$ such that $|G_1| = ... = |G_K| = 20$. The distributions are chosen to be isotropic, and the ratio between the lowest and the highest standard deviation is of 1 to 10. We draw points of a $\mathbb{R}^p$ space in two different scenarii. In $(\mathcal{S}_1)$, for a given dimension space $p = 500$ and a fixed isotropic noise level, we report the algorithm's performances as the signal-to-noise ratio $\Delta^2(\boldsymbol{\mu})/\sigma^2$ is increased from 1 to 15. In $(\mathcal{S}_2)$ we impose a fixed signal to noise ratio and observe the algorithm's decay in performance as the space dimension $p$ is increased from $10^2$ to $10^5$ (logarithmic scale). All reported points of the simulated space represent a hundred simulations, and indicate a median value with asymmetric standard deviations in the form of errorbars.

Solving for estimator $\widehat{B}^{corr}$ is a hard problem as $n$ grows. For this task we implemented an ADMM solver from the work of Boyd *et al.* [4] with multiple stopping criterions including a fixed number of iterations of $T = 1000$. The complexity of the optimization is then roughly $O(Tn^3)$. For reference, we compare the recovering capacities of $\widehat{\mathcal{G}}^{corr}$, labeled 'pecok' in Figure 1 with other classical clustering algorithm. We chose three different but standard clustering procedures: Lloyd's K-means algorithm [13] with a thousand K-means++ initialization of [1] (although in scenario $(\mathcal{S}_2)$, the algorithm is too slow to converge as $p$ grows so we do not report it), Ward's method for Hierarchical Clustering [23] and the low-rank clustering algorithm applied to the Gram matrix, a spectral method appearing in McSherry [15]. Lastly we include the CORD algorithm from Bunea *et al.* [5].

We measure the performances of estimators by computing the adjusted mutual information (see for instance [22]) between the truth and its estimate. In the two experiments, the results of $\widehat{\mathcal{G}}^{corr}$ are markedly better than that of other methods. Scenario $(\mathcal{S}_1)$ shows it can achieve exact recovery with a lesser signal to noise ratio than its competitors, whereas scenario $(\mathcal{S}_2)$ shows its performances start to decay much later than the other methods as the space dimension is increased exponentially.

Table 1 summarizes the simulations in a different light: for different parameter value on each line, we count the number of experiments (out of a hundred) that had an adjusted mutual information score equal to 0.9 or higher. This accounts for exact recoveries, or approximate recoveries that reasonably reflected the underlying truth. In this table it is also evident that $\widehat{\mathcal{G}}^{corr}$ performs uniformly better, be it for exact or approximate recovery: it manages to recover the underlying truth much sooner in terms of signal-to-noise ratio, and for a given signal-to-noise ratio it will represent the truth better as the embedding dimension increases.

Lastly Table 1 provides the median computing time in seconds for each method over the entire experiment. $\widehat{\mathcal{G}}^{corr}$ comes with important computation times because $\widehat{\Gamma}^{corr}$ is very costly to compute. Our method is computationally intensive but it is of polynomial order. The solving of a semidefinite program is a vastly developing field of Operational Research and even though we used the classical ADMM method of [4] that proved effective, this instance of the program could certainly have seen a more profitable implementation in the hands of a domain expert. All of the compared methods have a very hard time reaching high sample sizes $n$ in the high dimensional context.

The PYTHON3 implementation of the method used is found in open access here: martinroyer/pecok [18]

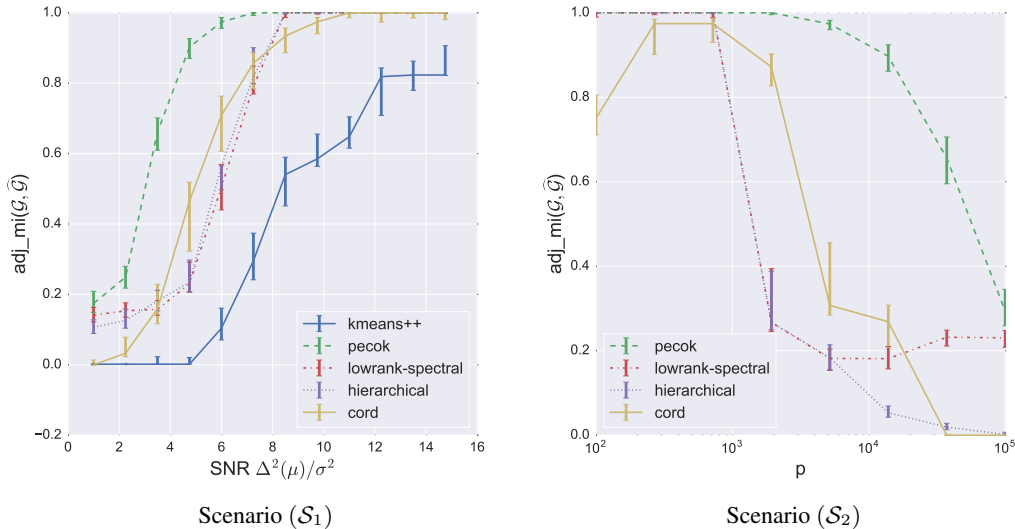

|  | Scenario ($\mathcal{S}_1$) | Scenario ($\mathcal{S}_2$) |

Figure 1: Performance comparison for clustering estimators and $\widehat{\mathcal{G}}^{corr}$, labeled 'pecok4' in reference to [6]. The adjusted mutual information equals 1 when the clusterings are identical, 0 when they are independent.

|  |  | hierarchical | kmeans++ | lowrank-spectral | pecok4 | cord |
|---|---|---|---|---|---|---|
| | 90% SNR=4.75 | 0 | 0 | 0 | 51 | 0 |
| | 90% SNR=6 | 0 | 0 | 0 | 100 | 0 |
| ($\mathcal{S}_1$) | 90% SNR=7.25 | 18 | 0 | 12 | 100 | 26 |
| | 90% SNR=8.5 | 100 | 0 | 100 | 100 | 76 |
| | med time (s) | 0.01 | 2.76 | 0.23 | 1.84 $(+18.92)^1$ | 0.76 |
| | 90% dim=$10^2$ | 100 | / | 100 | 100 | 94 |
| | 90% dim=$10^3$ | 0 | / | 0 | 100 | 31 |
| ($\mathcal{S}_2$) | 90% dim=$5.10^3$ | 0 | / | 0 | 100 | 0 |
| | 90% dim=$10^4$ | 0 | / | 0 | 49 | 0 |
| | med time (s) | 0.14 | $\infty$ | 0.19 | 1.94 $(+68.12)^1$ | 0.68 |

Table 1: Approximate recovery result for experiment ($\mathcal{S}_1$) and ($\mathcal{S}_2$): number of experiments that had a score superior to 90%, out of a hundred, and computing times over the experiments

# 6  Conclusion

In this paper we analyzed a new semidefinite positive algorithm for point clustering within the context of a flexible probabilistic model and exhibit the key quantities that guarantee non-asymptotic exact recovery. It implies an essential bias-removing correction that significanty improves the recovering rate in the high-dimensional setup. Hence we showed the estimator to be near-minimax, adapted to an effective dimension of the problem. We also demonstrated that our estimator can be optimally adapted to a data-driven choice of $K$, with a single tuning parameter. Lastly we illustrated on high-dimensional experiments that our approach is empirically stronger than other classical clustering methods. The $\widehat{\Gamma}^{corr}$ correction step of the algorithm, it can be interpreted as an independent, denoising step for the Gram matrix, and we recommend using such a procedure where the probabilistic framework we developed seems appropriate.

In practice, it is generally more realistic to look at approximate clustering results, but in this work we chose the point of view of exact clustering for investigating theoretical properties of our estimator. Our experimental results provide evidence that this choice is not restrictive, i.e. that our findings translate very well to approximate recovery. We expect our results to hold with similar speeds for approximate clustering, up to some logarithmic terms. One could think of adapting works on community detection by Guédon and Vershynin [12] based on Grothendieck's inequality, or work by Fei and Chen [10] from the stochastic-block-model community on similar semidefinite programs. In fact, referring to a detection bound by Banks, Moore, Vershynin, Verzelen and Xu (2017) [3], our only margin for improvement on the separation speed is to transform the logarithmic factor $\sqrt{\log n}$ into $\sqrt{\log K}$ when the number of clusters $K$ is of order $O(\log n)$ – otherwise the problem is rather open.

As for the robustness of this procedure, a few aspects are to be considered: the algorithm we studied solves for a convexified objective, therefore its performances are empirically more stable than that of an objective that would prove non-convex, especially in the high-dimensional context. In this work we also benefit from a permissive probabilistic framework that allows for multiple deviations from the classical gaussian cluster model, and come at no price in terms of the performance of our estimator. Points from a same cluster are allowed to have significantly different means or fluctuations, and the results for exact recovery with high probability are unchanged, near-minimax and adaptive. Likewise on simulated data the estimator proves the most efficient in exact as well as approximate recovery.

### Acknowledgements

This work is supported by a public grant overseen by the French National research Agency (ANR) as part of the "Investissement d'Avenir" program, through the "IDI 2015" project funded by the IDEX Paris-Saclay, ANR-11-IDEX-0003-02. It is also supported by the CNRS PICS funding HighClust. We thank Christophe Giraud for a shrewd, unwavering thesis direction.

## Footnotes

[1] The median time in parenthesis is the time to compute $\widehat{\Gamma}^{corr}$, as opposed to the main time for performing the SDP. Indeed the $\widehat{\Gamma}^{corr}$ is very time consuming, its cost is roughly $O(n^4 p)$. It must be noted that much faster alternatives, such as the one presented in [6], perform equally well (there is *no* significant difference in performance) for the recovery of $\mathcal{G}$, but this is outside the scope of this paper.

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
