[Supplementary Material · pc_nips_supp.pdf]

# Supplementary Material for Adaptive Clustering through Semidefinite Programming

**Martin Royer**
Laboratoire de Mathématiques d'Orsay, Univ. Paris-Sud, CNRS,
Université Paris-Saclay,
91405 Orsay, France
martin.royer@math.u-psud.fr

## A  Intermediate results

**Generic controls for exact recovery**

Let $\widehat{\Gamma}$ be any estimator of $\Gamma$ and let $\widehat{B} := \arg\max_{B \in \mathcal{C}_K} \langle \widehat{\Lambda} - \widehat{\Gamma}, B \rangle$.

**Theorem A.1.** *For $c_1, c_2 > 0$ absolute constants suppose that $|\widehat{\Gamma} - \Gamma|_V \leqslant \bar{\gamma}_n^2$ with probability $1 - c_1/n$, and that*

$$m\Delta^2(\boldsymbol{\mu}) \geqslant c_2\Big(\sigma^2(n + m\log n) + \mathcal{V}^2(\sqrt{n + m\log n}) + \bar{\gamma}_n^2 + \delta^2(\sqrt{n} + m)\Big), \qquad \text{(A.1)}$$

*then we have $\widehat{B} = B^*$ with probability larger than $1 - c_1/n$*

In the case where the number of groups is unknown we study $\widetilde{B} := \arg\max_{B \in \mathcal{C}} \langle \widehat{\Lambda} - \widehat{\Gamma}, B \rangle - \widehat{\kappa} \operatorname{tr}(B)$ for $\widehat{\kappa} \in \mathbb{R}$.

**Theorem A.2.** *For $c_3, c_4, c_5 > 0$ absolute constants suppose that $|\widehat{\Gamma} - \Gamma|_\infty \leqslant \bar{\gamma}_n^2$ with probability $1 - c_3/n$. Suppose that* (A.1) *is satisfied and that the following condition on $\widehat{\kappa}$ is satisfied*

$$c_4\Big(\mathcal{V}^2\sqrt{n} + \sigma^2 n + \bar{\gamma}_n^2 + \delta^2\sqrt{n}\Big) < c_5\widehat{\kappa} < m\Delta^2(\boldsymbol{\mu}), \qquad \text{(A.2)}$$

*then we have $\widetilde{B} = B^*$ with probability larger than $1 - c_3/n$*

**Concentration of random subgaussian Gram matrices**

A key result in our proof is the following concentration bound on the Gram matrix of centered, subgaussian, independent random variables.

**Lemma A.1.** *For some absolute constant $c_* > 0$, for $a \in [n]$ let $E_a$ be centered, independent random vectors in $\mathbb{R}^d$, $E_a \sim subg(\Sigma_a)$. Let $\mathbf{E} := \left[ \begin{smallmatrix} \cdots \\ E_a^T \\ \cdots \end{smallmatrix} \right] \in \mathbb{R}^{n \times d}$ then $\forall t \geqslant 0$*

$$\mathbb{P}\left[ \left| \mathbf{E}\mathbf{E}^T - \mathbb{E}\left[\mathbf{E}\mathbf{E}^T\right] \right|_{op} \geqslant 2\max_{a \in [n]} |\Sigma_a|_F \sqrt{t} + 2\max_{a \in [n]} |\Sigma_a|_{op} t \right] \leqslant 9^n 2 e^{-c_* t}. \qquad \text{(A.3)}$$

## B  Main proofs

### B.1  Proof of Proposition 1: identifiability

Suppose that $X_1, ..., X_n$ are $(\mathcal{G}, \boldsymbol{\mu}, \delta)$-clustered with $|\mathcal{G}| = K$, and $\rho(\mathcal{G}, \boldsymbol{\mu}, \delta) > 4$. Then we remark that for $(a, b) \in [n]^2$, $a \overset{\mathcal{G}}{\sim} b$ is equivalent to $|\nu_a - \nu_b|_2 \leqslant 2\delta$ because:

- if $a \overset{\mathcal{G}}{\sim} b$ then there exist $k \in [K]$ such that $|\nu_a - \nu_b|_2 \leqslant |\nu_a - \mu_k|_2 + |\mu_k - \nu_b|_2 \leqslant 2\delta$

- if $a \overset{\mathcal{G}}{\not\sim} b$ then there exist $(k, l) \in [K]^2$ such that $|\nu_a - \nu_b|_2 \geqslant |\mu_k - \mu_l|_2 - |\nu_a - \mu_k|_2 - |\nu_b - \mu_l|_2 > 4\delta - 2\delta > 2\delta$.

Now suppose there exist $\mathcal{G}'$ such that $X_1, ..., X_n$ are $(\mathcal{G}', \boldsymbol{\mu}', \delta')$-clustered with $|\mathcal{G}'| = K$ and $\rho(\mathcal{G}', \boldsymbol{\mu}', \delta') > 4$. By symmetry we can assume $\delta' \leqslant \delta$, and the previous remark shows that $\mathcal{G}'$ is a sub-partition of $\mathcal{G}$, ie $\mathcal{G}$ preserves the structure of $\mathcal{G}'$. But since $|\mathcal{G}| = |\mathcal{G}'|$ this implies $\mathcal{G} = \mathcal{G}'$. □

## B.2 Exact recovery with high probability

The proof for Theorem 1 (respectively Theorem 2) is a composition of Theorem A.1 (respectively Theorem A.2) and Proposition .

In this section, under Hypothesis (1), we have $\forall k \in [K], \forall a \in G_k : X_a \sim \mathrm{subg}(\Sigma_a)$. For $k \in [K]$, we define $\sigma_k^2 := \max_{a \in G_k} |\Sigma_a|_{op} \leqslant \sigma^2, \mathcal{V}_k^2 := \max_{a \in G_k} |\Sigma_a|_F \leqslant \mathcal{V}^2, \gamma_k^2 := \max_{a \in G_k} \mathrm{tr}(\Sigma_a) \leqslant \gamma^2$.

A number of proofs in this section are adapted from the proof ensemble of [1]. In it the authors use a latent model for variable clustering. A comparable model in this work would require to impose the following conditions on $X_1, ..., X_n$: identically distributed variables within a group (implying $\delta = 0$) and isovolumic, Gaussian distributions.

### B.2.1 Proof of Theorem A.1

In this theorem we only need to consider $B \in \mathcal{C}_K$, but the proof of Theorem A.2 is similar to this one, hence we will start by considering the more general $B \in \mathcal{C}$ and use $B \in \mathcal{C}_K$ at a later stage of the proof. Thus we want to prove that under some conditions, with high probability:

$$\langle \widehat{\Lambda} - \widehat{\Gamma}, B^* - B \rangle > 0 \text{ for all } B \in \mathcal{C} \setminus \{B^*\} \tag{B.1}$$

For $(a, b) \in G_k \times G_l$ for $(k, l) \in [K]^2$, let:

$$(S_1)_{ab} := -|\mu_k - \mu_l|_2^2/2 \tag{B.2}$$
$$(W_1)_{ab} := \langle \nu_a - \mu_k, \nu_b - \mu_l \rangle$$
$$(W_2)_{ab} := \langle \mu_k - \nu_a + \nu_b - \mu_l + E_b - E_a, \mu_k - \mu_l \rangle$$
$$(W_3)_{ab} := \langle E_b - E_a, \nu_a - \mu_k + \mu_l - \nu_b \rangle$$
$$(W_4)_{ab} := (\langle E_a, E_b \rangle - \Gamma_{ab})$$
$$(W_5)_{ab} := (\Gamma - \widehat{\Gamma})_{ab}$$

**Lemma B.1.** *Proving* (B.1) *reduces to proving*

$$\langle S_1 + W_1 + W_2 + W_3 + W_4 + W_5, B^* - B \rangle > 0 \text{ for all } B \in \mathcal{C} \setminus \{B^*\}. \tag{B.3}$$

The proof for Lemma B.1 is found in section B.2.3. So we need only concern ourselves with the quantities $S_1, W_1, W_2, W_3, W_4, W_5$. The term $S_1$ contains our uncorrupted signal and since $\langle S_1, B^* \rangle = 0$ it writes:

$$\langle S_1, B^* - B \rangle = \sum_{1 \leqslant k \neq l \leqslant K} \frac{1}{2} |\mu_k - \mu_l|_2^2 |B_{G_k G_l}|_1 \tag{B.4}$$

The other parts are noisy and must be controlled. The term $W_2$ is a simple subgaussian form controlled through the following lemma, proved in section B.2.4:

**Lemma B.2.** *For $c_2' > 0$ absolute constant, with probability greater than $1 - 1/n$:*

$$\forall B \in \mathcal{C}, \quad |\langle W_2, B^* - B \rangle| \leqslant \sum_{1 \leqslant k \neq l \leqslant K} \left( 2\delta + \sqrt{c_2'(\log n)(\sigma_k^2 + \sigma_l^2)} \right) |\mu_k - \mu_l|_2 |B_{G_k G_l}|_1. \tag{B.5}$$

To control the other noisy terms we now introduce a deterministic result:

**Lemma B.3.** *For any symmetric matrix $W \in \mathbb{R}^{n \times n}$ we have:*

$$\forall B \in \mathcal{C}, \quad |\langle W, B^* - B \rangle| \leqslant 6 |B^* W|_\infty \sum_{1 \leqslant k \neq l \leqslant K} |B_{G_k G_l}|_1$$

$$+ |W|_{op} \Big[ \sum_{1 \leqslant k \neq l \leqslant K} |B_{G_k G_l}|_1 / m + (\mathrm{tr}(B) - K) \Big]. \quad \text{(B.6)}$$

The proof for Lemma B.3 will be found in [1], p.21-22 until eq. (58).

As $B^* 1 = 1$ and $B^* \geqslant 0$, $|B^* W|_\infty \leqslant |W|_\infty$ so we use the lemma on terms $W_1$ and $W_3$ by bounding $|W|_\infty$ and $|W|_{op}$: for the term $W_1$ we use $|W_1|_\infty \leqslant \delta^2$ so $|W_1|_{op} \leqslant \delta^2 \sqrt{n}$. To control the term $W_3$, we use the subgaussian tail bound of (B.25) with $|\nu_a - \mu_k + \mu_l - \nu_b|_2 \leqslant 2\delta$ and a union bound over $(a, b) \in [n]^2$. We get that for $c'_3 > 0$ absolute constant, with probability greater than $1 - 1/n$, $|W_3|_\infty \leqslant \sqrt{c'_3 (\log n) \sigma^2 \delta^2}$ and $|W_3|_{op} \leqslant \sqrt{c'_3 (\log n) \sigma^2 \delta^2} \times \sqrt{n}$ therefore with probability greater than $1 - 1/n$, $\forall B \in \mathcal{C}$:

$$|\langle W_1, B^* - B \rangle| \leqslant \delta^2 \Big[ \sum_{1 \leqslant k \neq l \leqslant K} |B_{G_k G_l}|_1 (6 + \frac{\sqrt{n}}{m}) + \sqrt{n} (\mathrm{tr}(B) - K)_+ \Big] \quad \text{(B.7)}$$

$$|\langle W_3, B^* - B \rangle| \leqslant \sqrt{c'_3 (\log n) \sigma^2 \delta^2} \Big[ \sum_{1 \leqslant k \neq l \leqslant K} |B_{G_k G_l}|_1 (6 + \frac{\sqrt{n}}{m}) + \sqrt{n} (\mathrm{tr}(B) - K)_+ \Big] \quad \text{(B.8)}$$

For the term $W_4$ we introduce the following lemma, proved in section B.2.5:

**Lemma B.4.** *For $c'_4, c''_4 > 0$ absolute constants, with probability larger than $1 - 2/n$:*

$$\forall B \in \mathcal{C}, \quad |\langle W_4, B^* - B \rangle| \leqslant \Big[ 6 c'_4 (\mathcal{V}^2 \sqrt{\log n} + \sigma^2 \log n) / \sqrt{m} +$$

$$c''_4 (\mathcal{V}^2 \sqrt{n} + \sigma^2 n) / m \Big] \sum_{1 \leqslant k \neq l \leqslant K} |B_{G_k G_l}|_1 + (\mathrm{tr}(B) - K)_+ c''_4 (\mathcal{V}^2 \sqrt{n} + \sigma^2 n).$$

$$\text{(B.9)}$$

Lastly as the term $W_5$ is diagonal we have $|W_5|_{op} = |W_5|_\infty$ and $|B^* W_5|_\infty \leqslant |W_5|_\infty / m$ therefore:

$$\forall B \in \mathcal{C}, \quad |\langle W_5, B^* - B \rangle| \leqslant |W_5|_\infty \Big[ \frac{7}{m} \sum_{1 \leqslant k \neq l \leqslant K} |B_{G_k G_l}|_1 + (\mathrm{tr}(B) - K)_+ \Big] \quad \text{(B.10)}$$

Using those controls of $W_1, W_2, W_3, W_4, W_5$, in combination in a union bound in (B.3) we get for $c'_1 > 0$ absolute constant, with probability greater than $1 - c'_1/n$: $\forall B \in \mathcal{C}$,

$$\langle S_1 + W_1 + W_2 + W_3 + W_4 + W_5, B^* - B \rangle \geqslant \sum_{1 \leqslant k \neq l \leqslant K} \Big[ \frac{1}{2} |\mu_k - \mu_l|_2^2 -$$

$$\Big( 2\delta + \sqrt{2 c'_2 (\log n) \sigma^2} \Big) |\mu_k - \mu_l|_2 - (6 c'_4 \frac{\mathcal{V}^2 \sqrt{\log n} + \sigma^2 \log n}{\sqrt{m}} + c''_4 \frac{\mathcal{V}^2 \sqrt{n} + \sigma^2 n}{m})$$

$$- \frac{7}{m} |W_5|_\infty - (6 + \frac{\sqrt{n}}{m})(\delta^2 + \sqrt{c'_3 (\log n) \sigma^2 \delta^2}) \Big] |B_{G_k G_l}|_1$$

$$- (\mathrm{tr}(B) - K)_+ [c''_4 (\mathcal{V}^2 \sqrt{n} + \sigma^2 n) + (\delta^2 + \sqrt{c'_3 (\log n) \sigma^2 \delta^2}) \sqrt{n} + |W_5|_\infty] \quad \text{(B.11)}$$

We now use the fact that for this theorem we are only considering $B \in \mathcal{C}_K$, ie matrices such that $\mathrm{tr}(B) = K$ so we can discard the last line of (B.11). In this particular context we can improve the control provided by Lemma B.3 for $W_5$: as $\mathrm{tr}(B^*) = K$, we have for $\alpha \in \mathbb{R}$ : $|\langle W_5, B^* - B \rangle| \leqslant |\langle W_5 - \alpha I_n, B^* - B \rangle| + |\alpha (\mathrm{tr}(B) - K)|$. So by choosing $\alpha = (\max_a (W_5)_{aa} + \min_a (W_5)_{aa})/2$, we have $|W_5 - \alpha I_n|_{op} = |W_5 - \alpha I_n|_\infty = |W_5|_V / 2$ and therefore:

$$\forall B \in \mathcal{C}_K \quad |\langle W_5, B^* - B \rangle| \leqslant |W_5|_V \frac{7}{2m} \sum_{1 \leqslant k \neq l \leqslant K} |B_{G_k G_l}|_1 \quad \text{(B.12)}$$

In consequence we can replace $|W_5|_\infty$ by $|W_5|_V/2$ in the second line of (B.11), and with another union bound, by assumption we replace $|W_5|_V/2$ by $\bar\gamma_n^2/2$.

Lastly Lemma 3 p. 17 from [1] shows the only matrix in $\mathcal{C}_K$ whose support is included in $supp(B^*)$ is $B^*$, therefore $B \in \mathcal{C}_K \setminus \{B^*\}$ implies $\sum_{1 \leqslant k \neq l \leqslant K} |B_{G_k G_l}|_1 > 0$. Hence for $c_2 > 0$ absolute constant, the following condition on $\Delta(\boldsymbol{\mu})$ is sufficient to ensure exact recovery with probability larger than $1 - c_1/n$:

$$\Delta^2(\boldsymbol{\mu}) \geqslant c_2\big[\sigma^2 m \log n + \mathcal{V}^2\sqrt{m\log n} + \mathcal{V}^2\sqrt{n} + \sigma^2 n + \bar\gamma_n^2 + \delta^2(\sqrt{n}+m)\big] \times \frac{1}{m} \quad \text{(B.13)}$$

This concludes the proof for Theorem A.1. $\qquad\square$

### B.2.2 Proof of Theorem A.2: adaptive exact recovery

In this Theorem we need to take into account the additional penalization term $\widehat\kappa\,\mathrm{tr}(B)$. Notice it is equivalent to a correction by $\widehat\kappa I_n$ of our estimator $\widehat\Lambda - \widehat\Gamma$, therefore for $B \in \mathcal{C}$, $\langle\widehat\Lambda - \widehat\Gamma - \widehat\kappa I_n, B^* - B\rangle = \langle\widehat\Lambda - \widehat\Gamma, B^* - B\rangle + \widehat\kappa \times (\mathrm{tr}(B) - K)$. Therefore for Theorem A.2 we can follow the same proof as in Theorem A.1 until establishing (B.11), at which point we can use a union bound to use the assumption $|W_5|_\infty \leqslant \bar\gamma_n^2$. Consequently we have with probability greater than $1 - c_1'/n$: $\forall B \in \mathcal{C}$,

$$
\begin{aligned}
\langle S_1 + W_1 + W_2 + W_3 + W_4 + W_5, B^* - B\rangle \geqslant \sum_{1 \leqslant k \neq l \leqslant K} \Big[ & \frac{1}{2}|\mu_k - \mu_l|_2^2 \\
- \Big(2\delta + \sqrt{2c_2'(\log n)\sigma^2}\Big)|\mu_k - \mu_l|_2 - (6c_4' & \frac{\mathcal{V}^2\sqrt{\log n} + \sigma^2 \log n}{\sqrt{m}} + c_4''\frac{\mathcal{V}^2\sqrt{n} + \sigma^2 n}{m}) \\
- \frac{7}{m}\bar\gamma_n^2 - (6 + \frac{\sqrt{n}}{m})(\delta^2 + \sqrt{c_3'(\log n)\sigma^2\delta^2})\Big] & |B_{G_k G_l}|_1 \\
- (\mathrm{tr}(B) - K)_+[c_4''(\mathcal{V}^2\sqrt{n} + \sigma^2 n) + (\delta^2 + \sqrt{c_3'(\log n)\sigma^2\delta^2})\sqrt{n} + \bar\gamma_n^2] & + \widehat\kappa(\mathrm{tr}(B) - K)
\end{aligned}
$$

$$\text{(B.14)}$$

Using the assumption (A.1) of Theorem A.2 there exist $c_2' > 0$ such that with probability greater than $1 - c_1'/n$: $\forall B \in \mathcal{C}$,

$$
\begin{aligned}
\langle S_1 + W_1 + W_2 + W_3 + W_4, B^* - B\rangle \geqslant c_2'\Delta^2(\boldsymbol{\mu}) \sum_{1 \leqslant k \neq l \leqslant K} |B_{G_k G_l}|_1 \\
- (\mathrm{tr}(B) - K)_+[c_4''(\mathcal{V}^2\sqrt{n} + \sigma^2 n) + (\delta^2 + \sqrt{c_3'(\log n)\sigma^2\delta^2})\sqrt{n} + \bar\gamma_n^2] + \widehat\kappa(\mathrm{tr}(B) - K)
\end{aligned}
$$

$$\text{(B.15)}$$

From here, when $\mathrm{tr}(B) > K$, the left-hand side of (A.2) is sufficient to ensure recovery. When $\mathrm{tr}(B) = K$, we already established that $\sum_{1 \leqslant k \neq l \leqslant K} |B_{G_k G_l}|_1 > 0$ for all matrices $B \in \mathcal{C}_K \setminus \{B^*\}$ so (A.1) is sufficient in that case. Lastly note that $K - \mathrm{tr}(B) \leqslant \frac{1}{m}\sum_{1 \leqslant k \neq l \leqslant K} |B_{G_k G_l}|_1$ (see [1] eq. (57) p.21) so the right-hand side of (A.2) is sufficient condition for recovery when $\mathrm{tr}(B) - K < 0$. This concludes the proof of Theorem A.2. $\qquad\square$

### B.2.3 Proof of Lemma B.1

$$(\widehat{\Lambda} - \widehat{\Gamma})_{ab} = \langle X_a, X_b \rangle - \widehat{\Gamma}_{ab} = \langle \nu_a, \nu_b \rangle + \langle \nu_a, E_b \rangle + \langle \nu_b, E_a \rangle + \langle E_a, E_b \rangle - \widehat{\Gamma}_{ab} \tag{B.16}$$

$$= \langle \nu_a, \nu_b \rangle + \langle \nu_a - \nu_b, E_b - E_a \rangle + \langle \nu_a, E_a \rangle + \langle \nu_b, E_b \rangle + (W_4 + W_5)_{ab} \tag{B.17}$$

$$= \langle \nu_a, \nu_b \rangle$$
$$+ \langle \mu_k - \mu_l, E_b - E_a \rangle + (W_3)_{ab} + \langle \nu_a, E_a \rangle + \langle \nu_b, E_b \rangle + (W_4 + W_5)_{ab} \tag{B.18}$$

$$= -\langle \mu_k, \mu_l \rangle + \langle \nu_a - \mu_k, \nu_b - \mu_l \rangle + \langle \nu_a, \mu_l \rangle + \langle \mu_k, \nu_b \rangle$$
$$+ \langle \mu_k - \mu_l, E_b - E_a \rangle + (W_3)_{ab} + \langle \nu_a, E_a \rangle + \langle \nu_b, E_b \rangle + (W_4 + W_5)_{ab} \tag{B.19}$$

$$= -(S_1)_{ab} - \frac{1}{2}(|\mu_k|_2^2 + |\mu_l|_2^2) + (W_1)_{ab} + \langle \nu_a, \mu_l \rangle + \langle \mu_k, \nu_b \rangle$$
$$+ \langle \mu_k - \mu_l, E_b - E_a \rangle + (W_3)_{ab} + \langle \nu_a, E_a \rangle + \langle \nu_b, E_b \rangle + (W_4 + W_5)_{ab} \tag{B.20}$$

$$= -(S_1)_{ab} - \frac{1}{2}(|\mu_k|_2^2 + |\mu_l|_2^2) + (W_1)_{ab} + \langle \nu_a, \mu_k \rangle + \langle \mu_l, \nu_b \rangle$$
$$+ \langle \mu_k - \mu_l, \nu_b - \nu_a + E_b - E_a \rangle + (W_3)_{ab} + \langle \nu_a, E_a \rangle + \langle \nu_b, E_b \rangle + (W_4 + W_5)_{ab} \tag{B.21}$$

$$= -(S_1)_{ab} - \frac{1}{2}(|\mu_k|_2^2 + |\mu_l|_2^2) + (W_1)_{ab} + \langle \nu_a, \mu_k \rangle + \langle \mu_l, \nu_b \rangle$$
$$+ 2(S_1)_{ab} + (W_2)_{ab} + (W_3)_{ab} + \langle \nu_a, E_a \rangle + \langle \nu_b, E_b \rangle + (W_4 + W_5)_{ab} \tag{B.22}$$

Now since $(\langle \nu_a, \mu_k \rangle)_{(a,b) \in [n]^2} = (\langle \nu_a, \mu_k \rangle)_{a \in [n]} \times 1_n^T$, $(|\mu_k|_2^2)_{(a,b) \in [n]^2} = (|\mu_k|_2^2)_{a \in [n]} \times 1_n^T$, $(\langle \nu_b, \mu_l \rangle)_{(a,b) \in [n]^2} = 1_n \times (\langle \nu_b, \mu_l \rangle)_{b \in [n]}$, $(|\mu_l|_2^2)_{(a,b) \in [n]^2} = 1_n \times (|\mu_l|_2^2)_{b \in [n]}$, $(\langle \nu_a, E_a \rangle)_{(a,b) \in [n]^2} = (\langle \nu_a, E_a \rangle)_{a \in [n]} \times 1_n^T$, $(\langle \nu_b, E_b \rangle)_{(a,b) \in [n]^2} = 1_n \times (\langle \nu_b, E_b \rangle)_{b \in [n]}$ and since $B 1_n = B^* 1_n = (1_n^T B)^T = (1_n^T B^*)^T = 1_n$, we have:

$$\langle \widehat{\Lambda} - \widehat{\Gamma}, B^* - B \rangle = \langle S_1 + W_1 + W_2 + W_3 + W_4 + W_5, B^* - B \rangle \tag{B.23}$$

$\square$

### B.2.4 Proof of Lemma B.2: control of $|\langle W_2, B^* - B \rangle|$

By definition, $(W_2)_{ab} = 0$ when $k = l$ and $(B^*)_{ab} = 0$ when $k \neq l$ so we have $\langle W_2, B^* \rangle = 0$. Let $\langle A, B \rangle_{G_k G_l} = \sum_{(a,b) \in G_k \times G_l} A_{ab} B_{ab}$, we have:

$$\langle W_2, B^* - B \rangle = -\langle W_2, B \rangle = -\sum_{1 \leqslant k \neq l \leqslant K} \langle W_2, B \rangle_{G_k G_l} \leqslant \sum_{1 \leqslant k \neq l \leqslant K} |W_{2|G_k G_l}|_\infty |B_{G_k G_l}|_1 \tag{B.24}$$

Let $(a, b) \in G_k \times G_l$, we look at $(W_2)_{ab} = \langle E_b - E_a - (\nu_a - \mu_k) + (\nu_b - \mu_l), \mu_k - \mu_l \rangle = \langle E_a - E_b, \mu_k - \mu_l \rangle + \langle -(\nu_a - \mu_k) + (\nu_b - \mu_l), \mu_k - \mu_l \rangle$. The term on the right is a constant offset bounded by $2\delta |\mu_k - \mu_l|_2$. Let $z := \mu_k - \mu_l$, by Lemma C.1 $\langle E_a - E_b, z \rangle$ is a subgaussian variable with variance bounded by $(\sigma_k^2 + \sigma_l^2)|z|_2^2$ therefore its tails are characteristically bounded (see for example [4]), there exist $c_* > 0$ absolute constant such that $\forall t \geqslant 0$:

$$\mathbb{P}\left[ |\langle E_b - E_a, z \rangle| \geqslant |z|_2 \sqrt{\sigma_k^2 + \sigma_l^2} \times t \right] \leqslant e^{1 - c_* t^2} \tag{B.25}$$

This implies that $\forall t \geqslant 0, \mathbb{P}\left[ |(W_2)_{ab}| \geqslant |\mu_k - \mu_l|_2 (2\delta + \sqrt{\sigma_k^2 + \sigma_l^2} \times t) \right] \leqslant e^{1 - c_* t^2}$. We conclude with a union bound over all $(a, b) \in G_k \times G_l$, a union bound over all $(k, l) \in [K]^2$, $k \neq l$ and by taking $t = \sqrt{(1 + 3 \log n)/c_*}$. $\square$

### B.2.5 Proof of Lemma B.4: control of $|\langle W_4, B^* - B \rangle|$

Recall $(W_4)_{ab} = \langle E_a, E_b \rangle - \Gamma_{ab}$. We will prove Lemma B.4 by using the derivation of (B.6) combined with Lemma A.1 for control of the operator norm and the following lemma for the remaining part.

**Lemma B.5.** *For $c_4' > 0$ absolute constant, with probability greater than $1 - 1/n$:*

$$|B^* W_4|_\infty \leqslant c_4' \times (\mathcal{V}^2 \sqrt{\log n} + \sigma^2 \log n)/\sqrt{m}. \tag{B.26}$$

*Proof.* Let $(a,b) \in G_k \times G_l$, we rewrite $(B^* W_4)_{ab}$ as the sum of the following two terms:

$$(B^* W_4)_{ab} = \frac{u_b}{|G_k|} \times \mathbf{1}_{k=l} + \langle \widetilde{E}_k, E_b \rangle \text{ with } \begin{cases} u_b & := |E_b|_2^2 - \Gamma_{bb} \\ \widetilde{E}_k & := \frac{1}{|G_k|} \sum_{c \in G_k, c \neq b} E_c \end{cases} \tag{B.27}$$

The bound for $u_b$ uses Lemma C.3: $\forall t \geqslant 0 \ \mathbb{P}\left[ \big| |E_b|_2^2 - \mathbb{E}|E_b|_2^2 \big| \geqslant \mathcal{V}_l^2 \sqrt{t} + \sigma_l^2 t \right] \leqslant 2e^{-c_* t}$ so only the scalar product remains to be controlled. Notice that by Lemma C.1, $\sqrt{|G_k|}\widetilde{E}_k$ is a centered subgaussian with variance-bounding matrix $\widetilde{\Sigma} = \frac{1}{|G_k|}\sum_{c \in G_k, c \neq b} \Sigma_c$, therefore $|\widetilde{\Sigma}|_F \leqslant \mathcal{V}_k^2$ and $|\widetilde{\Sigma}|_{op} \leqslant \sigma_k^2$. So using Lemma C.3 again we find $\forall t \geqslant 0$:

$$\mathbb{P}\left[ 2|\sqrt{|G_k|}\langle \widetilde{E}_k, E_b \rangle| \geqslant \sqrt{2}\langle \widetilde{\Sigma}, \Sigma_b \rangle^{1/2}\sqrt{t} + |\widetilde{\Sigma}^{1/2}\Sigma_b^{1/2}|_{op} t \right] \leqslant 2e^{-c_* t} \tag{B.28}$$

Therefore using a union bound, then $\langle \widetilde{\Sigma}, \Sigma_b \rangle^{1/2} \leqslant \mathcal{V}_k \mathcal{V}_l \leqslant \mathcal{V}^2$ (Cauchy-Schwarz) and applying another union bound over all $(a,b) \in [n]^2$ with $t = (\log 4 + 3\log n)/c_*$ yields the result. $\quad\square$

We are ready to wrap-up the proof. From Lemma A.1 applied to $W_4$, taking $t = (\log 2 + n \log 9 + \log n)/c_*$ there exists $c_4'' > 0$ absolute constant such that we have with probability greater than $1 - 1/n$: $|W_4|_{op} \leqslant c_4''(\mathcal{V}^2 \sqrt{n} + \sigma^2 n)$. Now applying Lemma B.3 to $W_4$:

$$\begin{aligned} |\langle W_4, B^* - B \rangle| \leqslant & 6|B^* W_4|_\infty \sum_{1 \leqslant k \neq l \leqslant K} |B_{G_k G_l}|_1 \\ & + |W_4|_{op}\Big[ \sum_{1 \leqslant k \neq l \leqslant K} |B_{G_k G_l}|_1/m + (\mathrm{tr}(B) - K) \Big] \end{aligned} \tag{B.29}$$

Therefore combining the lemma with the derivations above and a union bound, we get with probability greater than $1 - 2/n$:

$$\begin{aligned} |\langle W_4, B^* - B \rangle| \leqslant & \Big[ 6c_4'(\mathcal{V}^2 \sqrt{\log n} + \sigma^2 \log n)/\sqrt{m} + c_4''(\mathcal{V}^2 \sqrt{n} + \sigma^2 n)/m \Big] \sum_{1 \leqslant k \neq l \leqslant K} |B_{G_k G_l}|_1 \\ & + (\mathrm{tr}(B) - K)_+ c_4''(\mathcal{V}^2 \sqrt{n} + \sigma^2 n) \end{aligned} \tag{B.30}$$

This concludes the proof for Lemma B.4. $\quad\square$

## B.3 Proof of Proposition 4, Gamma estimator $\widehat{\Gamma}^{corr}$

Let $a \in G_k$, $b_1 \in G_{l_1}$, $b_2 \in G_{l_2}$, using decomposition (1) and $2|xy| \leqslant x^2 + y^2$ we have for $a \in [n]$:

$$|\widehat{\Gamma}_{aa} - \Gamma_{aa}| = |\langle X_a - X_{b_1}, X_a - X_{b_2} \rangle - \Gamma_{aa}| \leqslant U_1 + \frac{3}{2}U_2 + 2U_3 + 3U_4 \tag{B.31}$$

$$\begin{aligned} \text{where:} \quad U_1 &:= \big| |E_a|_2^2 - \Gamma_{aa} \big| \\ U_2 &:= |\nu_a - \nu_{b_1}|_2^2 + |\nu_a - \nu_{b_2}|_2^2 \\ U_3 &:= \sup_{(b,c) \in [n]^2} \langle \frac{\nu_a - \nu_c}{|\nu_a - \nu_c|_2}, E_b \rangle^2 \\ U_4 &:= \sup_{(b,c) \in [n]^2, b \neq c} |\langle E_b, E_c \rangle| \end{aligned}$$

Control of $U_1 = \big| |E_a|_2^2 - \Gamma_{aa} \big|$: by using the first inequality from Lemma C.3 with $t = (2\log n + \log 2)/c_*$ there exists $c_1' > 0$ such that with probability greater than $1 - 1/n^2$:

$$U_1 \leqslant c_1' \times (\mathcal{V}_k^2 \sqrt{\log n} + \sigma_k^2 \log n) \tag{B.32}$$

Control of $U_3 = \sup_{(b,c) \in [n]^2} \langle \frac{\nu_a - \nu_c}{|\nu_a - \nu_c|_2}, E_b \rangle^2$: write $z = (\nu_a - \nu_c)/|\nu_a - \nu_c|_2$ and $Y = \Sigma_b^{-1/2} E_b \sim \mathrm{subg}(I_p)$ and $A = \Sigma_b^{1/2 \, T}(zz^T)\Sigma_b^{1/2}$, so that: $\langle z, E_b \rangle^2 = E_b^T zz^T E_b = Y^T A Y$.

Because $|z|_2 = 1$ and $zz^T$ is symmetric of rank 1 we have $|A|_F = |A|_{op} = \mathrm{tr}(A) \leqslant \sigma^2$ therefore we use Lemma C.2 with $t = (4\log n + \log 2)/c_*$ and then a union bound over all $(b,c) \in [n]^2$ so that with probability greater than $1 - 1/n^2$:

$$U_3 \leqslant c_3' \times \sigma^2 \log n \tag{B.33}$$

Control of $U_4 = \sup_{(b,c)\in[n]^2, b\neq c} |\langle E_b, E_c \rangle|$: using the fact that $E_b$ and $E_c$ are independent and the second inequality of Lemma C.3 with $t = (4\log n + \log 2)/c_*$, a union bound over all $(b,c) \in [n]^2$, there exists $c_4' > 0$ such that we have with probability greater than $1 - 1/n^2$:

$$U_4 \leqslant c_4' \times (\sigma^2 \log n + \mathcal{V}^2 \sqrt{\log n}) \tag{B.34}$$

Control of $U_2 = |\nu_a - \nu_{b_1}|_2^2 + |\nu_a - \nu_{b_2}|_2^2$: here we use the requirement that all groups are of length at least $m \geqslant 3$, there exist $(a_1, a_2) \in G_k \setminus \{a\}$, $(c,d) \in ([n] \setminus \{a, a_1, a_2\})^2$, let $Z = (X_c - X_d)/|X_c - X_d|_2$. For $a_u \in \{a_1, a_2\}$ we have $\langle X_a - X_{a_u}, Z \rangle = \langle \nu_a - \nu_{a_u}, Z \rangle + \langle E_a - E_{a_u}, Z \rangle$. By independence and Lemma C.1, $\langle E_a - E_{a_u}, Z \rangle$ is subgaussian with variance bounded by $2\sigma^2$. Therefore using the subgaussian tail bounds of (B.25) and a union bound, there exists $c_2' > 0$ absolute constant such that with probability over $1 - 1/n^2$: $V(a, a_1) \vee V(a, a_2) \leqslant 2\delta + c_2' \sigma \sqrt{\log n}$. Hence for $b_u \in \{b_1, b_2\}$ with probability over $1 - 1/n^2$:

$$|\langle X_a - X_{b_u}, X_c - X_d \rangle| \leqslant (2\delta + c_2' \sigma \sqrt{\log n})|X_c - X_d|_2 \tag{B.35}$$

Now suppose $l_1 \neq k$, choose $c \in G_k \setminus \{a\}, d \in G_{l_1} \setminus \{b_1\}$. We have $|X_c - X_d|_2 \leqslant |\mu_k - \mu_{l_1}|_2 + 2\delta + |E_c - E_d|_2$. We also have $\langle X_a - X_{b_1}, X_c - X_d \rangle = \langle \nu_a - \nu_{b_1} + E_a - E_{b_1}, \nu_c - \nu_d + E_c - E_d \rangle = \langle \mu_k - \mu_{l_1} + \delta_{ab} + E_a - E_{b_1}, \mu_k - \mu_{l_1} + \delta_{cd} + E_c - E_d \rangle$ for $\delta_{ab} = (\nu_a - \nu_{b_1}) - (\mu_k - \mu_{l_1})$ and $\delta_{cd} = (\nu_c - \nu_d) - (\mu_k - \mu_{l_1})$. Therefore:

$$|\langle X_a - X_{b_1}, X_c - X_d \rangle| \geqslant |\mu_k - \mu_{l_1}|_2^2/2 - 4\delta|\mu_k - \mu_{l_1}|_2 \tag{B.36}$$
$$- \frac{1}{2}\langle \frac{\mu_k - \mu_{l_1}}{|\mu_k - \mu_{l_1}|_2}, E_c + E_a - E_d - E_{b_1} \rangle^2 - 2 \sup_{(b,c,d)\in[n]^3} \langle \frac{\delta_{cd}}{|\delta_{cd}|_2}, E_b \rangle^2$$
$$- 4U_4 - 12\delta^2$$
$$\geqslant |\mu_k - \mu_{l_1}|_2^2/2 - 4\delta|\mu_k - \mu_{l_1}|_2 - 8U_3' - 2U_3'' - 4U_4 - 12\delta^2 \tag{B.37}$$

where $U_3' = \sup_{(b,l)\in[n]\times[K]} \langle \frac{\mu_k - \mu_l}{|\mu_k - \mu_l|_2}, E_b \rangle^2$, $U_3'' = \sup_{(b,c,d)\in[n]^3} \langle \frac{\delta_{cd}}{|\delta_{cd}|_2}, E_b \rangle^2$.

So combining the last derivations:

$$|\mu_k - \mu_{l_1}|_2^2/2 - 4\delta|\mu_k - \mu_{l_1}|_2 \leqslant (2\delta + c_2' \sigma \sqrt{\log n})(|\mu_k - \mu_{l_1}|_2 + 2\delta + |E_c - E_d|_2)$$
$$+ 8U_3' + 2U_3'' + 4U_4 + 12\delta^2 \tag{B.38}$$

Notice that $U_3', U_3''$ can be controlled exactly as $U_3$ was, and simultaneously: for $c_3'' > 0$ absolute constant, with probability greater than $1 - 1/n^2$: $8U_3' + 2U_3'' \leqslant c_3'' \sigma^2 \log n$.

We now control $|E_c - E_d|_2$: notice that by Lemma C.1, $E_c - E_d$ is $\mathrm{subg}(\Sigma_c + \Sigma_d)$. We have $\mathbb{E}\left[|E_c - E_d|_2^2\right] \leqslant \mathrm{tr}(\Sigma_c + \Sigma_d) \leqslant 2\gamma^2$, $|\Sigma_c + \Sigma_d|_F \leqslant 2\mathcal{V}^2 \leqslant 2\sigma\gamma$ and $|\Sigma_c + \Sigma_d|_{op} \leqslant 2\sigma^2$. Therefore by the first inequality of Lemma C.3 with $t = (4\log n + \log 2)/c_*$ and a union bound over all $(c,d) \in [n]^2$, there exists $c_2'' > 0$ absolute constant such that we have simultaneously with probability greater than $1 - 1/n^2$:

$$\sup_{(c,d)\in[n]^2} |E_c - E_d|_2 \leqslant c_2'' \sqrt{\gamma^2 + \sigma\gamma\sqrt{\log n} + \sigma^2 \log n} \leqslant c_2''(\gamma + \sigma\sqrt{\log n}) \tag{B.39}$$

Therefore with a union bound, with probability greater than $1 - 4/n^2$:

$$|\mu_k - \mu_{l_1}|_2^2/2 - (c_2' \sigma \sqrt{\log n} + 6\delta)|\mu_k - \mu_{l_1}|_2 \leqslant (2\delta + c_2' \sigma \sqrt{\log n})(2\delta +$$
$$(\gamma + \sigma\sqrt{\log n})(c_2'' + \frac{c_3''}{c_2'} + \frac{4c_4'}{c_2'})) + 12\delta^2$$
$$\tag{B.40}$$

Hence for $c_5' > 0$ absolute constant we have with probability greater than $1 - 4/n^2$: $|\mu_k - \mu_{l_1}|_2^2 \leqslant c_5'(\delta + \sigma\sqrt{\log n})(\delta + \sigma\sqrt{\log n} + \gamma)$. The same control can be derived simultaneously for $|\mu_k - \mu_{l_2}|_2^2$ by replacing $d \in G_{l_1} \setminus \{b_1\}$ by $d' \in G_{l_2} \setminus \{b_1, b_2\}$. We conclude that for $c_5'' > 0$ absolute constant, we have with probability greater than $1 - 4/n^2$:

$$U_2 \leqslant 2|\mu_k - \mu_{l_1}|_2^2 + 2|\mu_k - \mu_{l_2}|_2^2 + 16\delta^2 \leqslant c_5''(\delta + \sigma\sqrt{\log n})(\delta + \sigma\sqrt{\log n} + \gamma) \qquad (B.41)$$

Therefore with a union bound over all four terms $U_1, U_2, U_3, U_4$ and $a \in [n]$, for $c_6, c_7 > 0$ absolute constants we have with probability greater than $1 - c_6/n$: $|\widehat{\Gamma} - \Gamma|_\infty \leqslant c_7(\delta + \sigma\sqrt{\log n})(\delta + \sigma\sqrt{\log n} + \gamma)$. This concludes the proof of Proposition 4 $\qquad\square$

## B.4 Proof of Proposition 2

For this proof we rely heavily on the proof of Theorem A.1: let $\widehat{\Gamma} = 0$ so that $W_5 = \Gamma$, notice that $W_3$ and $W_4$ are centered. We take expectation of (B.3), therefore proving $\langle \Lambda + \Gamma, B^* - B \rangle > 0$ for all $B \in \mathcal{C}_K \setminus \{B^*\}$ is equivalent to proving:

$$\langle S_1 + W_1 + \mathbb{E}[W_2] + \Gamma, B^* - B \rangle > 0 \text{ for all } B \in \mathcal{C}_K \setminus \{B^*\} \qquad (B.42)$$

Notice that for $(a, b) \in G_k \times G_l$, $\mathbb{E}[(W_2)_{ab}] \leqslant 2\delta|\mu_k - \mu_l|_2$. Using this in combination with other arguments from the proof of Theorem A.1, that is using (B.4), (B.7) and (B.12), we have $\forall B \in \mathcal{C}_K$:

$$\langle S_1, B^* - B \rangle = \sum_{1 \leqslant k \neq l \leqslant K} \frac{1}{2}|\mu_k - \mu_l|_2^2|B_{G_k G_l}|_1 \qquad (B.43)$$

$$|\langle W_1, B^* - B \rangle| \leqslant \sum_{1 \leqslant k \neq l \leqslant K} \delta^2(6 + \frac{\sqrt{n}}{m})|B_{G_k G_l}|_1 \qquad (B.44)$$

$$|\langle \mathbb{E}[W_2], B^* - B \rangle| \leqslant \sum_{1 \leqslant k \neq l \leqslant K} 2\delta|\mu_k - \mu_l|_2|B_{G_k G_l}|_1 \qquad (B.45)$$

$$|\langle W_5, B^* - B \rangle| \leqslant \sum_{1 \leqslant k \neq l \leqslant K} \frac{7|\Gamma|_V}{2m}|B_{G_k G_l}|_1 \qquad (B.46)$$

Thus we have:

$$\langle S_1 + W_1 + \mathbb{E}[W_2] + W_5, B^* - B \rangle \geqslant \sum_{1 \leqslant k \neq l \leqslant K} \Big[\frac{1}{2}|\mu_k - \mu_l|_2^2 - 2\delta|\mu_k - \mu_l|_2$$
$$- \delta^2(6 + \frac{\sqrt{n}}{m}) - \frac{7|\Gamma|_V}{2m}\Big]|B_{G_k G_l}|_1 \qquad (B.47)$$

Hence we deduce that there exist $c_0$ absolute constant such that if $\rho^2(\mathcal{G}, \boldsymbol{\mu}, \delta) > c_0(6 + \sqrt{n}/m)$ and $m\Delta^2(\boldsymbol{\mu}) > 8|\Gamma|_V$, then we have $\arg\max_{B \in \mathcal{C}_K}\langle \Lambda + \Gamma, B \rangle = B^*$. Lastly as $B^*$ is in $\mathcal{C}_K^{\{0,1\}} \subset \mathcal{C}_K$, this concludes the proof. $\qquad\square$

## B.5 Proof of Proposition 3

Assume $X_1, ..., X_n$ is $(\mathcal{G}, \boldsymbol{\mu}, \delta)$-clustered with caracterizing matrix $B^*$ and define the following:

- $\delta = 0$ implying maximum discriminating capacity for $\mathcal{G}$ ie $\rho(\mathcal{G}, \boldsymbol{\mu}, \delta) = +\infty$.
- Let

$$B^* := \begin{bmatrix} \boxed{\frac{1}{m}} & & \\ & \boxed{\frac{1}{m}} & \\ & & \boxed{\frac{1}{m}} \end{bmatrix} \in \mathcal{C}_K^{\{0,1\}} \text{ and } B_1 := \begin{bmatrix} \boxed{2/m} & & \\ & \boxed{2/m} & \\ & & \boxed{\frac{1}{2m}} \end{bmatrix} \in \mathcal{C}_K^{\{0,1\}}$$

where $\boxed{\frac{1}{m}}$ represents constant square blocks of size $m$ and value $1/m$, and the other values in the matrices are zeros.

- $K = 3$ and for some $\Delta > 0$, $\mu_1 = (\Delta/\sqrt{2}, 0, 0)^T$ and $\mu_2 = (0, \Delta/\sqrt{2}, 0)^T$, $\mu_3 = (0, 0, \Delta/\sqrt{2})^T$ so that for $(a, b) \in G_k \times G_l$: $\Lambda_{ab} = \langle \mu_k, \mu_l \rangle = \Delta^2/2 \times \mathbf{1}\{a \overset{\mathcal{G}}{\sim} b\}$. Then $\Delta^2(\boldsymbol{\mu}) = \Delta^2$ and $\Lambda = (\Delta^2/2)mB^*$.

- For $\gamma_+ > \gamma_- > 0$ let $\Gamma = \text{diag}\,(\underbrace{\gamma_+, ..., \gamma_+}_{m}, \underbrace{\gamma_-, ..., \gamma_-}_{m}, \underbrace{\gamma_-, ..., \gamma_-}_{m})$

Then we have the following: $\langle B^*, \Gamma \rangle = \gamma_+ + 2\gamma_-$, $\langle B_1, \Gamma \rangle = 2\gamma_+ + \gamma_-$, $\langle B^*, \Lambda \rangle = \Delta^2/2 \times 3m$, $\langle B_1, \Lambda \rangle = \Delta^2/2 \times 2m$. Thus we have $\langle B^*, \Lambda + \Gamma \rangle < \langle B_1, \Lambda + \Gamma \rangle$ as soon as $m\Delta^2(\boldsymbol{\mu}) < 2(\gamma_+ - \gamma_-)$. This concludes the proof. $\qquad\square$

## C   Subgaussian properties and controls

**Lemma C.1.** $\forall a \in [n]$ let $Y_a \sim subg(\Sigma_a)$, independent, $\Sigma_a \in \mathbb{R}^{d \times d}$ then

$$Y = (Y_1^T, ..., Y_n^T)^T \sim subg(\text{diag}\,(\Sigma_a)_{a \in [n]}), \tag{C.1}$$

$$Z = \sum_{a \in [n]} c_a Y_a \sim subg(\sum_{a \in [n]} c_a^2 \Sigma_a). \tag{C.2}$$

*Proof.* By independence for $z = \{z_1^T, ..., z_n^T\}^T \in \mathbb{R}^{nd}$, $z_a \in \mathbb{R}^d$ we have

$$\mathbb{E}\left[e^{z^T(Y - \mathbb{E}Y)}\right] = \prod_{a=1}^n \mathbb{E}\left[e^{z_a^T(Y_a - \mathbb{E}Y_a)}\right] \leqslant \prod_{a=1}^n e^{z_a^T \Sigma_a z_a/2} = e^{z^T \,\text{diag}(\Sigma_a)_{a \in [n]} z/2}$$

$$\mathbb{E}\left[e^{z_1^T(Z - \mathbb{E}Z)}\right] = \prod_{a=1}^n \mathbb{E}\left[e^{z_1^T c_a(Y_a - \mathbb{E}Y_a)}\right] \leqslant \prod_{a=1}^n e^{z_1^T c_a^2 \Sigma_a z_1/2} = e^{z_1^T (\sum_{a \in [n]} c_a^2 \Sigma_a) z_1/2}$$

$\qquad\square$

**Lemma C.2.** *Hanson-Wright inequality for subgaussian variables*
*Let $Y$ be a centered random vector, $Y \sim subg(I_d)$, let $A$ be a matrix of size $d \times d$. There exists $c_* > 0$ such that for any $t \geqslant 0$*

$$\mathbb{P}\left[|Y^T A Y - \mathbb{E}\left[Y^T A Y\right]| \geqslant |A|_F \sqrt{t} + |A|_{op} t\right] \leqslant 2e^{-c_* t}. \tag{C.3}$$

*Proof.* A variation of the original Hanson-Wright inequality (Theorem 1.1 from [3]), it holds as $\sigma = 1$ bounds the subgaussian norm $|Y|_{\Psi_2} := \sup_{x \in \mathcal{S}_{d-1}} \sup_{p \geqslant 1} p^{-1/2}(\mathbb{E}|x^T Y|^p)^{1/p}$, a consequence of Lemma 5.5 from [4]. $\qquad\square$

**Lemma C.3.** *Subgaussian quadratic forms*
*Let $E, E'$ be centered, independent random vectors, $E \sim subg(\Sigma)$, $E' \sim subg(\Sigma')$, then for $t \geqslant 0$*

$$\mathbb{P}\left[||E|_2^2 - \mathbb{E}|E|_2^2| \geqslant |\Sigma|_F \sqrt{t} + |\Sigma|_{op} t\right] \leqslant 2e^{-c_* t} \tag{C.4}$$

$$\mathbb{P}\left[2|\langle E, E' \rangle| \geqslant \sqrt{2}\langle \Sigma, \Sigma' \rangle^{1/2} \sqrt{t} + |\Sigma^{1/2} \Sigma'^{1/2}|_{op} t\right] \leqslant 2e^{-c_* t}. \tag{C.5}$$

*Proof.* For the first inequality, we use Lemma C.2 with $Y = \Sigma^{-1/2} E$ and $A = \Sigma$. As for the second inequality, by Lemma C.1 we have $Y = (E^T \Sigma^{-1/2}, E'^T \Sigma'^{-1/2T})^T \sim subg(I_{2d})$. Then let us use Lemma C.2 with

$$A = \begin{pmatrix} 0 & \Sigma^{1/2} \Sigma'^{1/2} \\ \Sigma'^{1/2T} \Sigma^{1/2T} & 0 \end{pmatrix}$$

Notice that $|A|_F^2 = 2\langle \Sigma, \Sigma' \rangle$ and $|A|_{op} \leqslant |\Sigma^{1/2} \Sigma'^{1/2}|_{op}$ so the results follow. $\qquad\square$

**Proof of Lemma A.1: concentration of random subgaussian Gram matrices**.

Let $W := \mathbf{E}\mathbf{E}^T - \mathbb{E}[\mathbf{E}\mathbf{E}^T]$. Using the epsilon-net method as in Lemma 4.2 from [2], let $\mathcal{N}$ be a 1/4-net for $\mathcal{S}_{n-1}$ such that $|\mathcal{N}| \leqslant 9^n$ (see Lemma 5.2 [4]), we have for $u, v \in \mathcal{S}_{n-1}^2 : u^T W v \leqslant \max_{x \in \mathcal{N}} x^T W v + \frac{1}{4} \max_{u \in \mathcal{S}_{n-1}} u^T W v \leqslant \max_{x,y \in \mathcal{N}^2} x^T W y + \frac{1}{2} \max_{u,v \in \mathcal{S}_{n-1}^2} u^T W v$ hence

$$|W|_{op} \leqslant 2 \max_{x,y \in \mathcal{N}^2} x^T W y \quad \text{and} \quad \mathbb{P}\left[|W|_{op} \geqslant t\right] \leqslant \sum_{x,y \in \mathcal{N}^2} \mathbb{P}\left[x^T W y \geqslant t/2\right] \qquad (C.6)$$

Notice that this rewrites $x^T W y = \sum_{a=1}^n \sum_{b=1}^n x_a (E_a^T E_b - \Gamma_{ab}) y_b = (\sum_{a=1}^n E_a^T x_a)(\sum_{b=1}^n E_b^T y_b)^T - \mathbb{E}(\sum_{a=1}^n E_a^T x_a)(\sum_{b=1}^n E_b^T y_b)^T$. For $x, y \in \mathcal{N}^2$, let $x \otimes \Sigma^{1/2} := (x_1 \Sigma_1^{1/2}, ..., x_n \Sigma_n^{1/2})^T \in \mathbb{R}^{np \times p}$ and $Y = (E_1^T \Sigma_1^{-1/2}, ..., E_n^T \Sigma_n^{-1/2})^T \in \mathbb{R}^{np \times 1}$ (by Lemma C.1 we have $Y \sim \text{subg}(I_{np})$). We have

$$x^T W y = Y^T (x \otimes \Sigma^{1/2})(y \otimes \Sigma^{1/2})^T Y - \mathbb{E}[Y^T (x \otimes \Sigma^{1/2})(y \otimes \Sigma^{1/2})^T Y] \qquad (C.7)$$

Now define $A := (x \otimes \Sigma^{1/2})(y \otimes \Sigma^{1/2})^T$: we have $|A|_{op} \leqslant \max_{a \in [n]} |\Sigma_a|_{op}$ because for $z \in \mathbb{R}^p$, $|(x \otimes \Sigma^{1/2})z|_2^2 = \sum_{b=1}^n x_b^2 |\Sigma_b^{1/2} z|_2^2 \leqslant \max_{a \in [n]} |\Sigma_a|_{op} |z|_2^2$. As for the Frobenius norm, by Cauchy-Schwarz: $|(x \otimes \Sigma^{1/2})(y \otimes \Sigma^{1/2})^T|_F^2 = \sum_{a=1}^n \sum_{b=1}^n x_a^2 y_b^2 |\Sigma_a^{1/2} \Sigma_b^{1/2}|_F^2 \leqslant \max_{a \in [n]} |\Sigma_a|_F^2$. Therefore using Lemma C.2 on $Y$ we have $\forall t \geqslant 0 : \mathbb{P}\left[|Y^T A Y - \mathbb{E}\left[Y^T A Y\right]| \geqslant \max_{a \in [n]} |\Sigma_a|_F \sqrt{t} + \max_{a \in [n]} |\Sigma_a|_{op} t\right] \leqslant 2e^{-ct}$. Hence in conjunction with (C.6) we conclude the proof. $\square$