[Reviews · NeurIPS 2017]

Reviewer 1



The paper proposes a semidefinite relaxation for K-means clustering with isotropic and non-isotropic mixtures of sub-gaussian distributions in both high and low-dimensions. The paper has a good technical analysis of the methods proposed in it. There are some supporting empirical experiments too. However, one thing would have been useful to see was how computation time scales with both p and n. The empirical experiments considered n=30, which is pretty small for many application situations. So, for a better picture, it would have been good to see some results on computation times. The paper is nicely written and the flow of arguments in the paper is relatively clear. There are some minor typos like in line 111 in definition of b2, the subscript should be b in [n]\{a, b1}. The paper extends an approach of semidefinite relaxation for K-means clustering to account for non-isotropic behavior of mixture components. The method proposed can work for high-dimensional data too. The method will be quite useful but possibly computationally heavy too. So, an idea about the computational resources necessary to implement this method would have been useful.

Reviewer 2



This paper is a nice contribution to the theory of SDP algorithms for clustering, inspired by some of the techniques arising in SDP algorithms for community detection in the stochastic block model. This paper adds a correction term to the usual semidefinite relaxation of k-means, in order to improve on the weakness of previous SDPs in the anisotropic case. The paper proves a detailed non-asymptotic recovery guarantee, and analyzes the asymptotic scaling in multiple regimes, comparing the resulting asymptotics to known phenomena in the stochastic block model. The numerical experiments are compelling, and the writing is clear and enjoyable. Perhaps a slight weakness is that, while the introduction motivates a need for robust algorithms for clustering, questions of robustness could use more discussion. In particular, owing to the minima and maxima in the statement of \hat \Gamma^{corr}, I wonder whether this correction compromises robustness properties such as insensitivity to outliers.

Reviewer 3



This paper extends the PECOK method (Bunea et al, arXiv:1606.05100) to the case of the usual clustering setting, where PECOK was originally developed for relational data. The approach is somewhat standard: first transform the data into a relational data, which in this case is just the gram matrix, and then apply the PECOK methodology. Thus the conceptual innovation is rather marginal. The main technical effort is to show exact clustering for sub-Gaussian error. Conceptually, exact clustering is not realistic for any purpose. Theoretically, the proof simply follows that of PECOK and union bound. An interesting part is the de-biasing of the gram matrix (Proposition A.1). However, this seems to rely on the strong separation condition which is in turn needed for exact clustering. The paper would be improved if the author(s) can work something out in the more practical regime of approximate clustering, e.g., with vanishing mis-clustering proportion. On line 89, the author(s) used the word "natural relaxation" -- is it possible to show that $\mathcal{C}_K$ is the convex hull of $\mathcal{C}_K^{0,1}$ ?